# CAR T-Based Therapies in Lymphoma: A Review of Current Practice and Perspectives

**DOI:** 10.3390/biomedicines10081960

**Published:** 2022-08-12

**Authors:** Semira Sheikh, Denis Migliorini, Noémie Lang

**Affiliations:** 1Department of Hematology, Universitätsspital Basel, 4031 Basel, Switzerland; 2Department of Oncology, Hôpitaux Universitaires de Genève, 1205 Geneva, Switzerland; 3Center for Translational Research in Oncohematology, University of Geneva, 1206 Geneva, Switzerland

**Keywords:** cellular therapy, chimeric antigen receptor, lymphoma

## Abstract

While more than half of non-Hodgkin lymphomas (NHL) can be cured with modern frontline chemoimmunotherapy regimens, outcomes of relapsed and/or refractory (r/r) disease in subsequent lines remain poor, particularly if considered ineligible for hematopoietic stem cell transplantation. Hence, r/r NHLs represent a population with a high unmet medical need. This therapeutic gap has been partially filled by adoptive immunotherapy. CD19-directed autologous chimeric antigen receptor (auto-CAR) T cells have been transformative in the treatment of patients with r/r B cell malignancies. Remarkable response rates and prolonged remissions have been achieved in this setting, leading to regulatory approval from the U.S. Food and Drug Administration (FDA) of four CAR T cell products between 2017 and 2021. This unprecedented success has created considerable enthusiasm worldwide, and autologous CAR T cells are now being moved into earlier lines of therapy in large B cell lymphoma. Herein, we summarize the current practice and the latest progress of CD19 auto-CAR T cell therapy and the management of specific toxicities and discuss the place of allogeneic CAR T development in this setting.

## 1. Introduction

Non-Hodgkin lymphomas (NHLs) account for 4% of all cancers and represent the seventh leading cause of cancer death in the United States [1] and eleventh worldwide [2]. Despite booming novel antineoplastic agent development, a significant number of NHL patients continue to succumb to their disease, experiencing rapidly progressive disease or early relapse.

Autologous CAR (auto-CAR) T cell therapy is an individualized technology that genetically modifies the patient’s own T lymphocytes to specifically eradicate malignant cells and has drastically changed the landscape of many hematological malignancies, especially B cell NHLs. Key components of commercially available CAR T cell products consist of a CD-19 antigen-specific domain, a bridging transmembrane glycoprotein coupled to a costimulatory domain such as 4-1BB or CD28, which potentiates T cell activation signaling and improves CAR T cell expansion and persistence [3]. The process of CAR T cell therapy includes several steps: leukapheresis, ex vivo engineering and expansion of CAR T cells, and administration of a lymphodepleting conditioning regimen followed by infusion of the CAR T cell product.

Four different commercially-available CD19 auto-CARs are currently approved by the U.S. Food and Drug Administration (FDA) for r/r lymphomas: axi-cel (KTE-019), tisa-cel (CTL019), liso-cel (JCAR017) and brexu-cel (KTE-X19) [4], based on results of ZUMA-1 [5], JULIET [6], TRANSCEND [7] and ZUMA-2 [8], respectively (Table 1). CD19 auto-CAR products slightly vary in their engineering and manufacturing processes. Their main characteristics are summarized in Table 1. Although initial auto-CAR T cell therapy development mainly focused on r/r aggressive LBCL, it has nowadays also been implemented in some indolent lymphoma subtypes. The efficacy and safety results of pivotal and randomized trials will be described in more detail within the next section.

## 2. Efficacy of Autologous CAR T Cell Therapy in Lymphoma

### 2.1. Aggressive Lymphoma

#### 2.1.1. Large B-Cell Lymphoma

Large B cell lymphoma (LBCL) is a heterogeneous group that includes several entities with variable molecular patterns and prognosis [9,10,11,12]. Frontline immunochemotherapy is curative for roughly two-thirds of LBCL patients [13]; however, those presenting with primary refractory disease or experiencing early relapse have a dismal prognosis, with only approximately a quarter of patients benefiting from subsequent lines of therapy [14,15].

Pivotal clinical trials in ≥2 lines

Three pivotal single-arm early phase trials conducted in r/r adult LBCL patients who received at least two prior lines of systemic therapy, ZUMA-1 [5,16], JULIET [6,17], TRANSCEND [7,18], led to the registration of their respective CD19 auto-CAR product. Characteristics of the CAR T cell products and the safety and efficacy results of these trials are summarized in Table 2.

All trials included a relatively similar proportion of advanced-stage patients with a comparable median of prior lines; however, some distinctions in patient selection have to be highlighted. Notably, TRANSCEND included a higher proportion of patients above 65 years (41%) compared to ZUMA-1 (24%) and JULIET (23%), whereas the latter included more patients relapsing after autologous stem cell transplantation (49%) compared to 21% of patients in ZUMA-1 and 33% in TRANSCEND. The percentage of high-grade B cell lymphoma with MYC rearrangement was, respectively, 6%, 17% and 13% in ZUMA-1, JULIET and TRANSCEND. On the other hand, the number of primary refractory patients was higher in ZUMA-1 (98%) versus 55% in JULIET and 67% in TRANSCEND. Furthermore, trials differed in the inclusion of histologic subtype, with primary mediastinal B cell lymphoma (PMBL) only included in ZUMA-1 and TRANSCEND, and inpatient access to bridging therapy (BT) which was only permitted in JULIET and TRANSCEND. Finally, the proportion of “infused/enrolled” patients was significantly different across trials, respectively 91%, 69% and 85% for ZUMA-1, JULIET and TRANSCEND, with a significantly higher drop-out rate seen in JULIET, likely due to an extended time from leukapheresis to auto-CAR delivery for tisa-cel (54 days) (Table 1). Manufacturing time has been recognized as being of paramount importance in this r/r setting, as patients may experience a rapid progression of their disease whilst awaiting the auto-CAR product.

Taken together, these differences make pivotal cross-trial comparison difficult. However, all trials yielded remarkable overall response rates (ORR) ranging from 53% to 74%, with complete response (CR) reached in 39% to 54% of patients. Moreover, 65% to 80% of responders were able to maintain their remission with long-term follow-up. Long-term progression-free survival (PFS) and overall survival (OS) rates are summarized in Table 2 [16,17,18]. In their 4-year updated analysis of ZUMA-1, Jacobson et al. reported a strong correlation between event-free survival (EFS) and OS, and suggest to use EFS as a surrogate end-point for future trial design [16]. Several studies have attempted to indirectly compare outcomes of pivotal trials, adjusting for variables such as baseline characteristics, BT, and time to leukapheresis, but heterogeneity in the study design and limitations of data availability make it difficult to draw any conclusions, and head-to-head trials are needed [19,20,21,22]. A range of factors may affect CAR-T cell therapy efficacy, including patient and disease characteristics, CAR-T cell manufacturing and the type and depth of lymphodepletion. Attempts to identify molecular biomarkers of response to CAR T cell therapy (e.g., tumour expression of CD19, CD3, PD-1, PD-L1, CD3, TIM3 and LAG3) have so far been disappointing [6]; however, in the era of precision medicine, identifying patients more likely to respond to adoptive T-cell therapy and improving prognostic predictions is of paramount importance and should be prioritized for future trials.

Adverse events (AE) grade ≥3 were seen in 79–98% of patients, including 12–28% grade ≥3 infections. All grade cytokine release syndrome (CRS), as graded by Lee criteria [23], occurred in 42% to 92% of patients, including 2–11% of grade ≥3 CRS. Tocilizumab, corticosteroids and vasopressors were administered in 18–43%, 2–27% and 3–13% of cases, respectively. All grade neurological events, nowadays known as immune effector cell-associated neurotoxicity syndrome (ICANS), occurred in 30–67% of patients, with 10–32% reported as grade ≥3. Late cytopenias grade ≥3 were observed in 32–38%; immunoglobulin supplementation was necessary in 21–31% of cases. No new safety signals were reported with extended follow-up. Further details on the management of specific CAR T cell-related AEs are provided in Section 3.

Real-world evidence and outpatient setting

Several hundreds of patients have now been treated with auto-CAR T cells worldwide. Multiple groups retrospectively assessed the real-world outcomes and confirmed the feasibility and safety of this strategy [24,25,26,27,28,29,30,31,32,33,34] 8 August 2022 12:13:00 PM. Overall, patients treated in the standard of care setting tended to be older, with a third to half over the age of 65 years, and had a lower performance status and more advanced disease with a higher International Prognostic Index (IPI) score; approximatively half of these patients would not have been eligible for pivotal trials. Some real-world cohorts also reported a higher proportion of high-grade B cell lymphoma with MYC rearrangements [25,26,27]. Additionally, 53% to 84% of patients received BT, a factor shown to be predictive of reduced survival in retrospective analyses [35,36] and that may reflect a higher tumor burden and/or more aggressive disease at baseline.

Except from the UK experience [32], efficacy was surprisingly not impacted by less stringent patient selection; the best ORR ranged from 64% to 70% for axi-cel [24,25,27,29,32] and 46% to 62% for tisa-cel [26,27,29,30,31,32]. CR rates also remained consistent, ranging from 52% to 64% for axi-cel [24,25,27,29,31,32] and 38% to 44% for tisa-cel [26,27,30,31,32]. As previously demonstrated in pivotal studies, durability of response was sustained in complete responders. Approximately 10% of patients did not receive CAR infusion, either because of rapidly progressive disease or manufacturing failures (2–3%). Leukapheresis-to-infusion time was shorter for axi-cel (21 to 38 days) [24,25,27,29,31,32] than tisa-cel (32 to 46 days) [26,27,29,30,31,32]. Due to its later approval, such “real-world” data are presently lacking for liso-cel. Real-world safety results were consistent with those obtained in clinical trials, confirming a specific but manageable toxicity profile, with a tendency to a lower severe AE rate compared to pivotal trials, a finding potentially explained by increasing experience with CAR T cell toxicity management and earlier use of tocilizumab [37,38,39]. 

Finally, the OUTREACH multicenter phase 2 trial investigated the feasibility of liso-cel in the outpatient setting. The outcomes and safety of 52 patients receiving CAR T cell therapy as outpatients and monitored in non-university medical centers have recently been reported as similar to that in the inpatient setting. Of importance, nearly one-third of patients in this study did not require hospitalization [40].

Randomized clinical trials in earlier lines of therapy

Based on the outstanding results of pivotal trials, axi-cel, tisa-cel and liso-cel were tested against standard of care salvage chemotherapy followed by autologous stem cell transplantation (SOC) in three large multicenter phase 3 randomized trials, respectively: ZUMA-7 [41], BELINDA [42] and TRANSFORM [43]. Eligible patients had to have progressive disease or relapse within 12 months from initial immunochemotherapy completion. All trials evaluated EFS as a primary end-point, although the definition of EFS slightly varied from one trial to another, with ZUMA-7 and TRANSFORM including the start of a new treatment line as an event (Table 3). Contrary to BELINDA and TRANSFORM, ZUMA-7 did not permit patients to cross over to the CAR T arm. No new safety signals were reported across trials.

Two of these trials demonstrated the superiority of CD 19 auto-CAR T cell therapy over SOC: ZUMA-7 and TRANSFORM, whereas BELINDA failed to meet its primary end-point. Overall, baseline patient characteristics (age, disease stage, ECOG) were similar compared to prior pivotal trials (Table 3). ORR and CR rates were 83% and 65%, 46% and 28%, and 86% and 66% for ZUMA-7, BELINDA and TRANSFORM, respectively. EFS was significantly longer for axi-cel (HR 0.40, *p* < 0.0001) and liso-cel (HR, 0.35; *p* < 0.0001), while tisa-cel did not perform better than SOC (HR, 1.07; *p* = 0.61). Median EFS was 8.3, 3 and 10.1 months for axi-cel, tisa-cel and liso-cel versus 2, 3 and 2.3 months for SOC, respectively. Results for SOC were comparable to those reported in the literature [14,44,45], with only 35%, 33% and 46% of patients proceeding to autologous stem transplantation in each trial, respectively. Interestingly, quality of life assessed by patient-reported outcomes also favored axi-cel and liso-cel CAR T cells over SOC [46,47].

A few differences may have contributed to these discrepant results between studies: ZUMA-7 had the most stringent study design, with no BT permitted except for corticosteroids, thereby potentially excluding patients with high tumor burden and/or rapidly progressive disease. By contrast, most patients on TRANSFORM (63%) and BELINDA (83%) received BT, including 12% of patients receiving two different regimens of BT in the latter, potentially reflecting a population with a higher disease burden. Additionally, the median time to infusion was again longer for tisa-cel (52 days). Whether these differences have an impact on outcomes remains unclear as no direct prospective comparison of these CAR T cell products is yet available.

Primary and secondary CNS involvement

TRANSCEND was the only pivotal trial to allow patients with secondary CNS involvement (SCNSL), accounting for only 3% (N = 7) of patients. Small retrospective series evaluated outcomes of axi-cel and tisa-cel in SCNSL patients in the real-world setting [48,49,50] and were recently summarized in a systematic review (N = 44) [51]. No additional neurologic AEs were reported, and response rates seem similar to patients without CNS involvement, whereas the duration of response appears less sustained, although small patient numbers may limit the interpretability of results. Even though these findings require prospective confirmation, they confirmed the feasibility of CAR T cell therapy in this setting. Likewise, Frigault et al. recently reported the outcomes of 12 primary CNS lymphoma (PCNSL) patients treated with tisa-cel (NCT02445248), of which 6 achieved CR and maintained their remission at 1 year of follow-up [52]. Feasibility and safety were also confirmed by another group in 5 PCNSL patients [53]. The utility of liso-cel in PCNSL is currently being investigated in TRANSCEND WORLD (NCT03484702).

#### 2.1.2. Mantle Cell Lymphoma (MCL)

Two recent multicenter trials have shown clear clinical benefits from brexucabtagene autoleucel (brexu-cel) and liso-cel in r/r MCL. ZUMA-2 enrolled 74 patients with heavily pretreated r/r MCL, the vast majority failing or relapsing after BTKi [54]. The impressive ORR of 85% and CR of 59% seen in the intention-to-treat analysis led to FDA approval of brexu-cel for this indication (Table 1 and Table 2). Minimal residual disease (MRD), assessed by clonoSEQ (10^−6^ level), was undetectable in 79% of evaluable patients (N = 19) at 6 months and sustained after 3 years of follow-up. The benefit was seen across all high-risk subgroups, including BTKI refractoriness, high MIPI score, early progressors (POD24) and elevated Ki67 ≥ 50%. Due to the small number of patients, no conclusions could be made for TP53 mutated and blastoid variants, but these may have a less favorable outcome. Similarly, the first results of the MCL TRANSCEND cohort (N = 32) do compare favorably, with an ORR of 84% and over half of patients in CR [55].

Preliminary real-world data of two multicenter groups have also emerged, one from the U.S. and one from Europe [56,57], both reproducing safety and efficacy results of ZUMA-2 with the best ORR and CR rates in the range of 86–91% and 64–79%, respectively. In the U.S. study, the 3-month PFS rate was 80.6%, and the 6-month OS rate was 82.1% [56], while the 12-month PFS and OS rates of the European study were 76% and 61%, respectively [57]. Manufacturing failures occurred in 6–8% of cases, and 65–82% of patients received BT compared to 32% in ZUMA-2. Finally, an ongoing clinical trial is exploring the efficacy of tisa-cel in combination with ibrutinib in patients with r/r MCL (TARMAC, NCT04234061). Even though it is too early to draw conclusions on the curative potential of CAR T cells in this setting, this modality offers durable responses in over half of this poor-prognosis population. 

#### 2.1.3. T-Cell Lymphoma

T-cell lymphoproliferative disorders constitute a highly heterogenous group of lymphomas related to poor outcomes and an unmet need for r/r patients or ineligible for transplantation. The applicability of CAR T cell therapy in T cell lymphoma is much more challenging; limitations have been well-described by Safarzadeh et al. in their recent review and include the lack of T-cell tumor-specific targetable antigens (CD3, CD5, CD7) with an inherent risk of CAR T-mediated T-cell aplasia, CAR T cell fratricide resulting in poor CAR T persistence and the risk of malignant T cell contamination during leukapheresis resulting in a malignant auto-CAR construct, among others [58]. To our knowledge, only a few clinical results have been published. A recent phase 1 study reported a promising safety profile and high response rates (19/20 CR in the bone marrow by day 28, 5/9 extramedullary CR) with a CD7-targeted CAR in 20 patients with r/r T-cell acute lymphoblastic leukemia/lymphoma [59]. Other ongoing early phase trials are currently evaluating the safety and efficacy profile of CAR T cells directed against CD7 (NCT04840875, NCT04689659, NCT04480788, NCT05059912, NCT04599556, NCT03690011, NCT04823091), CD5 targeted CAR T (NCT04594135, NCT03081910, NCT05138458) and CD4-targeted CAR (NCT03829540). Other CAR modalities, such as allogeneic T and NK CAR constructs, are also under investigation (NCT04984356, NCT02742727).

### 2.2. Indolent Lymphoma

Despite significant improvements in the armamentarium of novel therapeutic strategies within the past decade, most indolent lymphomas remain incurable, with the exception of rare patients eligible for allogeneic stem cell transplantation (allo-SCT) [60]. Indolent lymphomas are highly heterogeneous [61], with certain subgroups having high-risk clinical and molecular features that may result in a more aggressive disease course and significantly reduced survival [62]. These patients also tend to have shorter response duration with subsequent lines of therapies, and thus, the management of patients who develop acquired resistance remains challenging [63].

#### 2.2.1. Follicular Lymphoma and Marginal Zone Lymphoma

The efficacy of CAR T cells was first demonstrated in a heavily pretreated advanced-stage FL patient in 2010 [64], and this was followed by a small case series [65,66]. Axi-cel has been recently approved by the FDA in this setting based on interim results of ZUMA-5, evaluating the benefit of axi-cel in r/r indolent lymphoma [67] (Table 1 and Table 2). This phase 2 trial enrolled 124 FL and 16 marginal zone lymphoma patients. Among the 80 evaluable FL patients, 94% responded, including 79% of CR. With 31 months of follow-up, the 18-month PFS and OS rates were 65.6% and 88%, respectively [68]. These results also compare favorably to the outcomes of a retrospective multicenter cohort treated with standard immunochemotherapy [69]. Tisa-cel has also been evaluated in high-risk FL. With a median follow-up of 17 months, 69% of the 97 patients infused in the ELARA study achieved a CR, with an ORR of 86% [70]. At the time of writing, no results of the TRANSCEND FL cohort have so far been published.

#### 2.2.2. Chronic Lymphocytic Leukemia and Small Lymphocytic Lymphoma

Early reports described the preliminary activity of CAR T cell therapy in chronic lymphocytic leukemia/small lymphocytic lymphoma (CLL/SLL), either as monotherapy or in combination with BTKi [71,72,73,74,75]. TRANSCEND CLL-004 is a Phase 1 study that enrolled 22 patients withr/r CLL/SLL who were treated with liso-cel. Half of patients were refractory to both BTKi and BCL2i, 83% had high-risk genetic features. ORR and CR rates were 82% and 45%, respectively. In the 20 evaluable subjects, MRD was undetectable in 75% and 65% in the blood and bone marrow, respectively [76]. Liso-cel and ibrutinib in combination were deemed safe and tested in 19 r/r CLL/SLL patients. ORR and CR/CRi were 98% and 48% at one month post-CAR infusion. MRD negativity was 89% in the blood and 79% in the bone marrow, assessed, respectively, by flow cytometry and next-generation sequencing (10–4 level) [77]. The ongoing ZUMA-8 Phase 1/2 trial is currently investigating the role of brexu-cel in this setting [78]. Additionally, promising activity has also been described in patients with transformed CLL/SLL (Richter syndrome) [72,79,80,81]. 

### 2.3. Hodgkin Lymphoma

CAR T cell development in Hodgkin Lymphoma has so far been less promising than in B cell NHL [82]; however, Ramos et al. recently demonstrated encouraging safety and efficacy results using a fludarabine-based lymphodeleting regimen in 41 heavily pretreated HL patients with 7 median prior lines of therapy [83]. Among the 32 evaluable patients, ORR was 72%, and 59% of patients achieved CR; 1-year PFS and OS were 36% and 94%, respectively. CD30 CAR T has also been evaluated in combination with a PD1 inhibitor in a Phase 2 trial conducted in 12 CD30 positive lymphoma patients (9 HL, 2 grey zone and 1 angioimmunoblastic lymphoma) with some durable responses [84].

## 3. CAR T Cell Associated Toxicities

Due to the nature of CAR T cells as “living” cellular drugs, they display a unique toxicity profile that is distinct from that usually expected with standard chemotherapy regimens. In addition, as CAR T cell therapy is moving towards earlier lines of treatment and is being more broadly employed in the lymphoma treatment landscape, optimal management strategies of associated side effects are of high relevance. CRS, ICANS, and late cytopenias constitute key challenges in the treatment of lymphoma patients with CAR T cells.

### 3.1. Cytokine Release Syndrome (CRS)

CRS is the most commonly observed CAR T cell-associated toxicity [23,85,86,87,88,89,90]. CRS is a supraphysiologic inflammatory state triggered by inflammatory cytokines and chemokines, including interferon (IFN)-y and tumor necrosis factor (TNF)-a, released by CAR T cells after engaging with the target antigen CD19; resultant activation of bystander host antigen-presenting cells (APCs) and T cells has been shown to be the primary source of IL-6 and IL-1, both identified as key drivers of CRS [23,85]. Multiple other inflammatory substances have also been identified as contributing factors in the hyperinflammatory process, which in a feedforward loop activates the vascular endothelium, resulting in further release of IL-6 [86,87,91,92,93], and an imbalance of endothelial homeostatic factors can result in a loss of vascular integrity, hemodynamic instability, capillary leak and consumptive coagulopathy [86,91,94]. 

Clinically, CRS typically presents with fever, associated with non-specific constitutional symptoms such as fatigue, myalgia, and anorexia. Symptoms can progress on a continuum, resulting in tachycardia, hypotension and/or hypoxia. If unchecked, patients may further deteriorate with symptoms of respiratory failure, organ dysfunction and shock [94,95,96]. CRS usually occurs within hours to days after CAR T cell infusion but rarely presents beyond 14 days after therapy. The median time to onset of CRS varies depending on the specific CAR T cell product and disease-associated factors. For example, for tisa-cel comprising the 4-1BB (CD137) costimulatory domain, the associated CRS risk peaks at day 7 after CAR T cell administration for patients with DLBCL, whereas patients treated with axi-cel, containing the CD28 costimulatory domain, usually experience CRS earlier, at 2 days after CAR T cell administration [6,91,97]. In the absence of established biomarker profiles that can reliably predict a patient’s individual CRS risk, patients need to be carefully followed and assessed for hallmarks of evolving CRS.

#### 3.1.1. CRS Definition and Severity

Because initial clinical trials used different grading systems to characterize and assess the severity of CRS, comparisons of safety profiles of different CAR T cell products and evaluation of differences in reported CRS incidence are limited. As the clinical experience with CAR T cell products continues to evolve, there have been several efforts to update and harmonize grading criteria for CRS in clinical trials. In 2019, the American Society for Transplantation and Cellular Therapy (ASTCT) published consensus guidelines that simplified CRS grading, using fever (defined as temperature ≥ 38 °C) that is temporally associated with CAR T cell administration (within 24 h to 3 weeks) as the prerequisite for CRS diagnosis. Hypotension and hypoxia are used as the principal determinants of CRS grade and severity, while organ toxicities no longer contribute to CRS grading. It is anticipated that the ASTCT CRS grading criteria will be widely implemented in clinical trials and in routine patient care going forward and, therefore, will allow for an objective and reproducible assessment of CRS in the clinical setting [37]. 

#### 3.1.2. CRS Management

There are currently a number of guidelines available for CRS management [37,38,39]. The mainstay of CRS management comprises symptomatic measures (including antipyretics and intravenous fluids) predominantly for lower grade CRS, as well as anti-cytokine therapy (e.g., with tocilizumab) and corticosteroids. As CRS may evolve as a continuum, individual treatment decisions should be made at the bedside according to the clinical judgement of the treating physician. CRS-mediated fever is a diagnosis of exclusion. Concurrent conditions such as underlying infection should be considered, and a symptom-oriented diagnostic work-up should be carried out. Careful observation of patients for CRS and early intervention to reverse symptoms have significantly reduced the rates of high-grade CRS and ICU admissions for patients undergoing CAR T cell therapy [37,38,39].

### 3.2. Neurotoxicity/ICANS

Neurotoxicity, or ICANS, is another common but usually self-limited and reversible side effect. In contrast to CRS, the pathophysiology driving neurotoxicity is not well understood. Several reports have implicated endothelial activation and disruption of the blood-brain barrier (BBB), which may facilitate the influx of cytokines into the CNS, but elevated levels of the excitatory NMDA receptor agonists glutamate and quinolinic acid have also been described [98,99,100,101].

Clinical manifestations of ICANS can be diverse, ranging from tremor and dysgraphia to expressive dysphasia and encephalopathy, as well as seizures and cerebral edema, which can be fatal. Atypical presentations, including quadriparesis and acute leukoencephalopathy, have also been reported [98,99,102]. 

Pivotal studies of CAR T cell therapy in lymphoma reported ICANS Grade ≥3 in 10% to 32% of patients, although grading schemes have varied between different clinical trials [6,7,97]. A biphasic pattern of neurotoxicity has been observed, with the first phase often occurring in the context of CRS, a median time to onset of symptoms of 5 days, and a median duration of 17 days [37,98,103,104]. A second, delayed phase of neurotoxicity has been reported in approximately 10% of patients, including cerebrovascular events and neurocognitive morbidity [105]. Risk factors associated with the development of neurotoxicity, its duration and severity include patient-related factors such as younger age, higher tumor burden and a history of early and/or high-grade CRS, as well as product-related characteristics such as CAR design and choice of lymphodepletion regimen [6,98,104].

#### 3.2.1. ICANS Grading

As for CRS, the ASTCT Consensus grading system is recommended for the grading of ICANS. The ICANS grading system incorporates the 10-point Immune Effector Cell-Associated Encephalopathy (ICE) score, a standardized screening tool for encephalopathy. Patients are graded according to the most severe symptom in five neurological domains. Non-specific neurological symptoms such as headache, tremors or hallucinations are excluded from ICANS grading [37].

#### 3.2.2. ICANS Management

Patients receiving CAR T cell therapy should be closely monitored for the development of neurotoxicity. ICANS is primarily a clinical diagnosis, and work-up of neurological symptoms by imaging, lumbar puncture or encephalogram may be required to rule in or exclude differential diagnoses. The role of antiepileptic prophylaxis has not as yet been conclusively determined for patients who receive CAR T cell therapy, but many patients receive prophylaxis [37,38]. For cases of ICANS not occurring in the context of CRS, corticosteroids are the first-line treatment; although dosing may vary based on neurologic symptoms, dexamethasone 10 mg IV every 6–8 h is usually administered. Short courses of steroids used for ICANS management have not been shown to affect response to CAR T cell therapy or progression-free survival. Tocilizumab has poor BBB penetration and therefore is only given together with corticosteroids for ICANS that develops concurrently with CRS; this constitutes a largely empirical approach as there are no clinical trial data that compare different treatment strategies [6,37,38].

### 3.3. Other Toxicities Associated with Auto-CAR T Cell CD19 Products

Depletion of normal CD19-expressing B cells due to “on-target, off-tumor” effects of CD19-directed CAR T cells may lead to prolonged B cell aplasia and profound immune deficiency after CAR T cell therapy, thereby posing a unique challenge for both acute and long-term prevention of infections [106,107]; select patients may benefit from immunoglobulin prophylaxis. 

Prolonged cytopenias of ≥Grade 3 and lasting beyond 28 days after CAR T cell infusion may occur in approximatively 20–40% of patients, both in clinical trials and in the real-world setting, underscoring the importance of careful follow-up of these patients. The mechanism for these prolonged cytopenias remains poorly understood and may be multifactorial. Some patients respond to growth factor support and corticosteroids, but future work is needed to understand the incidence, causation, and ramifications of cytopenias at various time points after CAR T cell administration [39,108,109].

Hemophagocytic lymphohistiocytosis (HLH) has been observed as a rare complication of CAR T cell therapy, with an incidence of approximately 3.5%. Diagnostically, HLH may present a challenge as there is an overlap of some of its presenting signs and symptoms with CRS; a high index of suspicion is required for prompt diagnosis, and both corticosteroids and anti-cytokine therapy with an IL-6 antagonist have been used in CAR T cell patients [38].

## 4. Allogeneic CAR T Cell Development and Activity in NHL

Emerging long-term follow-up data of auto-CAR T cell therapy in patients with lymphoma show that this approach is not curative in the majority of patients, with a combination of disease factors and product characteristics having been identified as possible causes for auto-CAR T cell failure. Approximately 40% of patients treated with auto-CAR T cell therapy may achieve long-term remission, but outcomes of patients who do not respond to or experience relapse post-CAR T-cells remain poor, with a reported median OS of 3.6 months [110]. In addition, with the increasing use of auto-CAR T cell therapy for patients with r/r lymphoma in the standard of care setting, it has become apparent that other limitations include lack of accessibility, delays in manufacturing, variable product quality, and an incomplete understanding of resistance mechanisms [111]. For example, patients with a high tumor burden or rapidly progressive disease were underrepresented in pivotal clinical trials. Resistance mechanisms described in auto-CAR T cell therapy include antigen-positive relapse, resulting from a specific CAR T cell phenotype affecting CAR T cell persistence and an immunosuppressive microenvironment, for example, and antigen-negative relapses where antigen loss may be driven by alternative splicing, epitope masking, antigen downregulation or lineage switching of the lymphoma [110,112]. Consequently, additional improvements in CAR T cells as a treatment modality are further warranted. Strategies currently under investigation in clinical trials include switching the target antigen, multi-CAR constructs, or allo-CAR T cells. 

Allogeneic CAR T cells (allo-CAR) may overcome some of these limitations; allo-CARs can be manufactured from healthy donor cells and could become available as an “off-the-shelf” product, increasing availability for patients and reducing the need for bridging chemotherapy [113,114]. Salient differences between auto-CAR and allo-CARs are summarized in Table 4.

A small study of 20 patients with relapsed or refractory B cell malignancies following allogeneic bone marrow transplant and/or donor lymphocyte infusion showed the safety and feasibility of this approach. Patients received a single infusion of donor-derived anti-CD19 CAR-T cells with no prior lymphodepletion chemotherapy. A total of 8 of the 20 patients achieved a response, and none of the patients developed GvHD [115].

Allo-CARs sourced from healthy donors for patients with r/r lymphomas include ALLO-501, a genetically modified anti-CD19 CAR T cell product where the TCR alpha gene is disrupted to reduce the risk of GvHD, and the CD52 gene is disrupted to permit the use of an anti-CD52 monoclonal antibody (mAb), ALLO-647, for selective and prolonged host lymphodepletion. The phase I ALPHA study (NCT03939026) treated patients with r/r lymphoma after ≥2 lines of therapy. At the last updated analysis, 46 of 47 enrolled patients had been treated with ALLO-501; the median time from enrollment to start of therapy was 5 days, and 20% of patients had received prior auto-CAR T cell therapy. ALLO-501 was safe and manageable, with no GvHD reported, no Gr ≥ 2 ICANS, and only limited CRS observed. Cytopenias were observed in 82.6% of patients, and ≥Grade 3 infections occurred in 23.9% of patients, which is similar to that observed in auto-CAR T trials. The 6-month CR rate was 36.4% for large cell lymphoma patients [116,117].

In the ongoing ALPHA2 study (NCT04416984), patients with r/r lymphoma receive either a single or consolidation dose of ALLO-501A, which uses Cellectis technologies to disrupt the T cell receptor alpha gene (TRAC) and the CD52 gene. Consolidation and single dosing had a comparable safety profile, and the efficacy profile was improved with consolidation dosing. Persistence of CAR T cells at D28 and expansion after the consolidation dose was observed, as well as deepening of responses in patients whose initial response was PR. Further follow-up is needed to assess the durability of the response [88,118].

Another allo-CAR T cell product, PBCAR0191, is being evaluated in patients with CD19+ r/r B cell NHL who have received ≥2 prior therapies (NCT03666000). Subjects receive either a standard (×3 days fludarabine and cyclophosphamide; sLD) or enhanced (×4 days fludarabine and ×3 days cyclophosphamide; eLD) lymphodepletion regimen preceding PBCAR0191 infusion. The median time from eligibility confirmation to PBCAR0191 infusion was 7 days. PBCAR0191 demonstrated dose- and LD-dependent cell expansion kinetics. Of 15 patients dosed, none experienced GvHD; there were no cases of Grade ≥3 CRS and 1 case of Grade 3 ICANS. The CR rate at day ≥28 ranged from 33% for patients receiving sLD to 80% receiving eLD. A total of 4 of 15 (27%) responding patients underwent allo-SCT. Duration of response assessment is ongoing [119].

CTX110 is an allo-CAR that uses CRISPR/Cas9 technology to insert the CAR construct into the TRAC locus to simultaneously disrupt endogenous TCR expression to reduce the risk of GvHD. In addition, major histocompatibility complex (MHC) expression is eliminated to avoid rejection of the CAR-T cells and thereby improve CAR-T cell persistence. A Phase 1/2 study in patients with r/r B cell malignancies is ongoing (NCT04035434); as of August 2021, data from 24 patients showed a comparable safety profile to auto-CAR T therapy and a 58% ORR and 36% CR rate [120]. 

As well as being sourced from healthy donors that have previously not been exposed to cytotoxic chemotherapy and therefore may have fitter and less exhausted T cells, another advantage of allo-CARs is that they can be created from T cell subsets that may confer properties such as memory or stemness, which could be associated with better persistence and influence long-term efficacy outcomes, and also from other cell types such as natural killer (NK), gamma-delta and induced pluripotent stem cells [112,113].

In summary, allo-CAR approaches may overcome some of the current limitations inherent in auto-CAR T cells. However, more follow-up is required to properly assess the long-term impact and potential consequences of gene editing, allo-immunization and GvHD on the safety, feasibility and efficacy of these products.

## 5. Conclusions and Future Outlook

In the last decade, auto-CAR T cells have transformed the treatment landscape and outlook of patients with r/r lymphoma, with pivotal clinical trials demonstrating high response rates and durable remissions and raising the possibility of cure in this difficult-to-treat patient population. Four auto-CAR T cell products have gained regulatory approval for the treatment of B cell malignancies and are available in the standard of care setting, with real-world data showing reproducible results. Auto-CAR T cell therapy is associated with unique toxicities, including CRS and ICANS, and the requirement for treatment centres to be experienced in managing these is currently a limiting factor to the widespread adoption of CAR T cell therapy. Future work is needed to identify predictors of response and improve the benefit/risk profile to minimize toxicity and treatment burden for patients. Novel CAR T cell constructs may reduce the risk of CRS and ICANS while also optimizing antigen recognition on lymphoma cells and CAR T cell persistence. Logistical and economic hurdles may be overcome by exploring off-the-shelf allo-CAR cells that do not require individualized manufacturing, can be cryopreserved and banked in batches and therefore have the potential to reduce lead times and create a more accessible platform for cell therapies, especially for patients with a high disease burden who require urgent therapy.

## Figures and Tables

**Table 1 biomedicines-10-01960-t001:** Currently FDA-approved CD19 auto-CAR T cell products.

CAR T Product	Year of Approval	Clinical Trial	Study Design	Patient Population	Engineering and Manufacturing Characteristics	Dose	Median Time from Leukapheresis to Product Release (Days)	Lympho-Depleting Regimen
Axi-cel (KTE-019, Yescarta)	2017	ZUMA-1 (NCT02348216)	Phase 2 single-arm, open-label, multicenter, international	LBCL ≥ 2 lines	CD28, retrovirus Fresh leukapheresis	2 × 10^6^ cells/kg (max. 2 × 10^8^ cells/kg)	17	Flu 30 mg/m^2^ + Cy 500 mg/m^2^ daily × 3d Flu 30 mg/m^2^ + Cy 500 mg/m^2^ daily × 3d
	2021	ZUMA-5 (NCT03105336)	Phase 2 single-arm, open-label, multicenter, international	FL ≥ 3 lines			17	
	2022 ^1^	ZUMA-7 (NCT03391466)	Phase 3 randomized, multicenter, international	LBCL ≥ 1 lines			13	
Brexu-cel (KTE-X19, Tecartus)	2020	ZUMA-2 (NCT02601313)	Phase 2 single-arm, open-label, multicenter, international	MCL ≥ 3 lines	CD28, retrovirus Fresh leukapheresis		16	
Tisa-cel (CTL019, Kymriah)	2018	JULIET (NCT02445248)	Phase 2 single-arm, open-label, multicenter, international	LBCL ≥ 2 lines	4-1BB, lentivirus Frozen leukapheresis	0.6–6 × 10^8^ cells	54	Flu 25 mg/m^2^ + Cy 250 mg/m^2^ daily × 3d or Be 90 mg/m^2^ daily × 2d
Liso-cel (JCAR017, Breyanzi)	2021	TRANSCEND (NCT02631044)	Phase 1 single-arm, open-label, multicenter, international	LBCL ≥ 2 lines	4-1BB, retrovirus Fresh leukapheresis	50–110 × 10^6^ cells (Separate infusions of CD4+/CD8+ CAR-T cells at 1:1 dose ratio)	24	Flu 30 mg/m^2^ + Cy 300 mg/m^2^ daily × 3d
	2022 ^1^	TRANSFORM (NCT03575351)	Phase 3 randomized, multicenter, international	LBCL ≥ 1 lines			26	

^1^ Approved as second line in large B cell lymphomas who are refractory to first-line chemoimmunotherapy or who experience disease relapse within 12 months of first-line chemoimmunotherapy. Abbreviations: Axi-cel (axicabtagene ciloleucel); Be, bendamustine; Brexu-cel (brexucabtagene autoleucel); Cy, cyclophosphamide; d, day; FL, follicular lymphoma; Flu, fludarabine; liso-cel (lisocabtagene maraleucel); LBCL, large B cell lymphomas; MCL, mantle cell lymphoma; tisa-cel (tisagenlecleucel).

**Table 2 biomedicines-10-01960-t002:** Characteristics and results of pivotal clinical trials for CD 19 auto-CAR T cell therapies approved in relapsed/refractory B cell lymphoma.

Variable	ZUMA-1 NCT02348216	JULIET NCT02445248	TRANSCEND NCT02631044	ZUMA-2 NCT02601313	ZUMA-5 ^1^ NCT03105336
Auto-CAR product	Axi-cel	Tisa-cel	Liso-cel	Brexu-cel	Axi-cel
Histologic type (%)	DLBCL (76), PMBL (8), tFL (16)	DLBCL (80), HGBL (15), tFL (18), Other (2)	DLBCL (51), HGBL (13), FL grade 3b (1), PMBL (6), tFL (22), tiNHL (7)	MCL	iNHL, including FL (84) and MZL (16)
Enrolled patients–no/Infused patients–no (%)	111/101 (91)	165/115 (69)	344/269 (85) ^2^	74/58 (92)	127/124 (98)
Median age, yr (range)	58 (23–76)	56 (27–76)	63 (18–86)	65 (38–19)	60 (34–79)
Bridging therapy (%patients)	Corticosteroids (NA)	Chemotherapy (93)	Chemotherapy (59)	Any (35)	Any (4)
Median prior lines of therapy (range)	3 (2–4)	3 (1–6)	3 (1–8)	3 (1–5) ^3^	3 (2–4) ^4^
Best overall response rate (%)	74	53	73	91	94
Complete response rate (%)	54	39	53	68	79
Median follow-up (mo)	51.1	40.3	29.3	35.6	30.9
Median duration of response (mo)	11.1	NE	23.1	38.6	NR
Median progression-free survival (mo)	5.9	2.9	6.8	39.6	NR
Progression-free survival at 24 mo (%)	40	35	40.6	52.9	65.6 (18 mo)
Progression-free survival among patients with CR at 24 mo (%)	70	80	49.5	71.8	NR
Median overall survival (mo)	25.8	11.1	27.3	NR	NR
Overall survival at 24 mo (%)	44 (48 mo)	45	50.5	~84	88 (18 mo)
Adverse Events grade ≥3 (%)	98	89	79	99	85
Serious Adverse Events(%)	48	65	45	68	46
**Adverse Events of special interest**					
Cytokine release syndrome (CRS) ^5^					
All (%)	92	58	42	91	78
Grade ≥3 (%)	11	17	2	15	6
Tocilizumab	43 ^7^	24	18 ^8^	59	50 (all iNHL)
Corticosteroids (%)	27 ^7^	16	2	22	18 (all iNHL)
Vasopressors (%)	13	10	3	16	5 (all iNHL)
Neurological events ^6^					
All (%)	67	20	30	63	56
Grade ≥3 (%)	32 ^7^	11	10	31	15
Tocilizumab	43 ^7^	20	NA	26	36
Corticosteroids (%)	27	12	NA	38	6
Infections grade ≥ 3 (%)	28	19	12	32	18 (all iNHL)
Late cytopenia grade ≥ 3 ^9^ (%)	38	32	37	26	33
Immunoglobulin (%)	31	33	21	32	9 (all iNHL)

Note: The purpose of this table is to summarize currently available data. Head-to-head studies have not been performed, and no comparisons can be made. ^1^ Results for the FL group if not indicated as iNHL. ^2^ Twenty-five patients received a product that failed to meet specifications but was deemed safe to administer. ^3^ Patients must be exposed to anthracyclines- or bendamustine-containing regimen, anti-CD20 and BTKi. ^4^ Patients must be exposed to prior anti-CD20 and alkylating agents. ^5^ Cytokine release syndrome in this table are all graded according to Lee scale criteria, even though CRS in JULIET was initially reported according to Penn grading scale. ^6^ Neurological events reported according to National Cancer Institute Common Terminology Criteria for Adverse Events (CTCAE version 4.03). ^7^ Received for either CRS and/or ICANS. ^8^ Tocilizumab alone was given to 10% of patients. ^9^ Cytopenias ≥ 28 days in JULIET, ≥30 days in the other studies. Abbreviations: Axi-cel (axicabtagene ciloleucel); Be, bendamustine; brexu-cel (Brexucabtagene autoleucel); BTKi, bruton tyrosine kinase inhibitor; CAR, chimeric antigen receptor; Cy, cyclophosphamide; CRS, cytokine release syndrome; DLBCL, diffuse large B cell lymphoma; FL, follicular lymphoma; Flu, fludarabine; HGBL, high-grade B cell lymphoma; iNHL, indolent non-Hodgkin lymphoma; liso-cel (Lisocabtagene maraleucel); LBCL, large B cell lymphomas; MCL, mantle cell lymphoma; MZL, marginal zone lymphoma; NA, not available or reported; NE, not estimated; NR, not reached; PMBL, primary mediastinal b cell lymphoma; tFL, transformed follicular lymphoma; tiNHL, transformed indolent non-Hodgkin lymphoma; tisa-cel (tisagenlecleucel).

**Table 3 biomedicines-10-01960-t003:** Characteristics and results of CD19 auto-CAR arm in randomized phase 3 trials in relapse/refractory B cell lymphoma ≥ 1 line of therapy.

Variable	ZUMA-7 NCT03391466	BELINDA NCT03391466	TRANSFORM NCT03575351
CAR product	Axi-cel	Tisa-cel	Liso-cel
Primary end-point definition (Event-free survival)	SD or PD up to day 150, new lymphoma treatment, death	SD or PD disease at week 12, death	SD or PD at week 9, new lymphoma treatment, death
Crossover (%)	Not permitted	Allowed (51)	Allowed (55)
Manufacturing success (%)	100	97	99
Lymphodepleting regimen	Flu 30 mg/m^2^ + Cy 500 mg/m^2^ daily × 3 days	Flu 25 mg/m^2^ + Cy 250 mg/m^2^ daily × 3 days ^1^	Flu 30 mg/m^2^ + Cy 300 mg/m^2^ daily × 3 days
Enrolled patients–no (assigned to CAR)	359 (180)	322 (162)	182 (92)
CAR-infused patients–no (%)	170 (94)	155 (96)	89 (97) ^2^
Median time to infusion (days)	13	52	36
Bridging therapy (%)	Corticosteroids only (36)	Chemotherapy (83)	Chemotherapy (63)
Histologic type (%)			
DLBCL (ABC subtype)	70 (9)	62 (32)	58 (23)
HGBL	17	24	24
PMBL	-	7	9
FL grade 3b	-	3	1
tiNHL	11	17	8
Other	13	3	1
Secondary CNS involvement	-	3	-
Median age, yr	58 (range 21–80)	59.5 (range 19–79)	60 (IQR 54–68)
Secondary IPI score ≥ 2 (%)	46	65	40
Refractory disease ^3,4^ (%)	74	66	45
Best overall response rate (%)	83	46	86
Complete response rate (%)	65	28	66
Median progression-free survival (mo)	14.5	NA	14.8
Progression-free survival (%)	~46 (24 mo)	NA	52 (12 mo)
Median event-free survival (mo)	8.3	3	10.1
Event-free survival (%)	40.5 (24 mo)	NA	45 (12 mo)
Median overall survival (mo)	NR	NR	NR
Overall survival at 24 months (%)	~61 (24 mo)	NA	~79 (12 mo)
Median follow-up (mo)	25	10	6.2
Adverse Event grade ≥ 3 (%)	91	84	92
Serious Adverse Events (%)	50	36	48
Adverse Events of special interest			
Cytokine release syndrome			
All (%)	92	61	49
Grade ≥ 3 (%)	6	5	1
Tocilizumab	64	52	23 ^5^
Corticosteroids (%)	24	17	-
Vasopressors (%)	6	NA	-
Neurological events ^6^			
All (%)	60	10	12
Grade ≥ 3 (%)	21	2	4
Corticosteroids (%)	32	NA	8 ^7^
Infections grade ≥ 3	14	NA	15
Cytopenia grade ≥ 3 (>30 days) ^8^	29	NA	43

Note: The purpose of this table is to summarize data. Head-to-head studies have not been performed and no comparisons can be made. ^1^ If contraindicated bendamustine 90 mg/m^2^ for 2 days. ^2^ One patient received a nonconforming CAR product. ^3^ Refractory disease defined as a lack of complete response to first-line therapy. ^4^ Cytokine release syndrom graded according to Lee scale criteria. ^5^ 13% associated to corticosteroids. ^6^ Neurological events reported according to National Cancer Institute Common Terminology Criteria for Adverse Events (CTCAE versio n 4.03). ^7^ 1% associated with tocilizumab. ^8^ Defined as >30 days persistent grade ≥ 3 cytopenia. Abbreviations: Axi-cel (axicabtagene ciloleucel); Be, bendamustine; brexu-cel (Brexucabtagene autoleucel); BTKi, bruton tyrosine kinase inhibitor; CAR, chimeric antigen receptor; CNS, central nervous system; Cy, cyclophosphamide; d, day; DLBCL, diffuse large b-cell lymphoma; FL, follicular lymphoma; Flu, fludarabine; HGBL, high-grade B-cell lymphoma; IPI, international prognostic index; iNHL, indolent non-hodgkin lymphoma; liso-cel (Lisocabtagene maraleucel); LBCL, large B-cell lymphomas; MCL, mantle cell lymphoma; MZL, marginal zone lymphoma; NA, not available; NR, not reached; PMBL, primary mediastinal b-cell lymphoma; tFL, transformed follicular lymphoma; tiNHL, transformed indolent non-hodgkin lymphoma; tisa-cel (tisagenlecleucel); yr, year.

**Table 4 biomedicines-10-01960-t004:** Differences between auto-CAR and allo-CAR T cell products.

Characteristic	Auto-CAR	Allo-CAR
Cell source and product	- Autologous patient-derived T cells - Heterogeneous product	- Healthy-donor derived - Standardized product
Manufacturing process	- Leukapheresis required - Transduction of (unselected) apheresed T cell product	- Healthy donor source - Pre-manufactured
Availability	- Depends on individualized manufacturing times, can range from 2–6 weeks - Commercially available products in standard of care setting	- Readily available “off the shelf” - Products in clinical trials, longer treatment follow up and pivotal trials pending
Side effects	- CRS - ICANS - Cytopenias - HLH	- CRS - ICANS - Cytopenias - Allo-immunization - Graft versus host disease - Rejection of allogeneic cells
Repeat dosing	Possible but may require repeat apheresis, under investigation	Possible and can consider alternative donor, under investigation
Persistence	Months to years	Weeks to months

## Data Availability

Not applicable.

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
