# Peer review of "CAR T-Based Therapies in Lymphoma: A Review of Current Practice and Perspectives"

_biomedicines, 2022, doi:10.3390/biomedicines10081960_

Round 1
Reviewer 1 Report
The work well understands and describes in an organized way the theme of the CAR T and the present studies. The tables that summarize the results in a comparative way are excellent.
However, in my opinion it is totally wrong to include mantle cells among indolent lymphomas. They must be separated from that arrangement and considered as aggressive lymphomas.
Furthermore, it is really too little to say a single sentence on the issue of CAR T in T lymphomas. It is necessary to expand this paragraph.
Author Response
Point-by-point responses to the Reviewer(s)' Comments to Author
Response to Reviewer #1
The work well understands and describes in an organized way the theme of the CAR T and the present studies. The tables that summarize the results in a comparative way are excellent.
However, in my opinion it is totally wrong to include mantle cells among indolent lymphomas. They must be separated from that arrangement and considered as aggressive lymphomas.
Furthermore, it is really too little to say a single sentence on the issue of CAR T in T lymphomas. It is necessary to expand this paragraph.
- We thank the Reviewer for suggesting to include mantle cell lymphomas among aggressive lymphomas, and have added the section on MCL to aggressive lymphoma (2.1.2. Mantle cell lymphoma (MCL), lines 254-279). We appreciate you found tables comprehensive. We considered your remark regarding the T-cell lymphoma which has been moved in the aggressive lymphoma section (2.1.3. T-cell lymphoma, lines 281-298) and extended it as follows:
“ T-cell lymphoproliferative disorders constitute a highly heterogenous group of lymphomas related with poor outcome and an unmet need for r/r patients or ineligible to transplantation. The applicability of CAR T cell therapy in T cell lymphoma is much more challenging; limitations have been well-described by Safarzadeh et al in their recent review and include the lack of T-cell tumor-specific targetable antigens (CD3, CD5, CD7) with and inherent of CAR-T-mediated T-cell aplasia, CAR T cell fratricide resulting in poor CAR T persistence and the risk of malignant T cell contamination during leukapheresis resulting in a malignant auto-CAR construct, among others[58]. To our knowledge, only few clinical results have been published. A recent phase 1 study reported a promising safety profile and high response rates (19/20 CR in the bone marrow by day 28, 5/9 extramedullary CR) with a CD7-targeted CAR in 20 patients with r/r T-cell acute lymphoblastic leukemia / lymphoma[59]. Other ongoing early phase trials are currently evaluating the safety and efficacy profile of CAR T cells directed against CD7 (NCT04840875, NCT04689659, NCT04480788, NCT05059912, NCT04599556, NCT03690011, NCT04823091), CD5 targeted CAR T (NCT04594135, NCT03081910, NCT05138458) and CD4 targeted CAR (NCT03829540). Other CAR modalities, such as allogeneic T and NK CAR constructs are also under investigation (NCT04984356, NCT02742727)”.
Reviewer 2 Report
The present review is well-written and complete, despite not becoming excessively long.
I have only a few minor concerns
1) Since MCL and T-cell lymphomas (at least the ones for which CAR-T are discussed) are definitely aggressive lymphomas, they should be included in a paragraph on Aggressive lymphomas with DLBCL and HGL.
2) I know the trials and the available data. However, the authors might dare to wrap up the few indications emerging (only in DLBCL/HGL) regarding biological prognostic factors (COO, translocations etc). If no evidence, even weak, did emerge, please discuss it as one major need. Identifying patients more prone to respond to a given treatment should be the MAIN objective of all trials in the era of precision medicine
Author Response
Point-by-point responses to the Reviewer(s)' Comments to Author
Response to Reviewer #2
The present review is well-written and complete, despite not becoming excessively long.
I have only a few minor concerns
1) Since MCL and T-cell lymphomas (at least the ones for which CAR-T are discussed) are definitely aggressive lymphomas, they should be included in a paragraph on Aggressive lymphomas with DLBCL and HGL.
2) I know the trials and the available data. However, the authors might dare to wrap up the few indications emerging (only in DLBCL/HGL) regarding biological prognostic factors (COO, translocations etc). If no evidence, even weak, did emerge, please discuss it as one major need. Identifying patients more prone to respond to a given treatment should be the MAIN objective of all trials in the era of precision medicine
We thank the Reviewer 2 for their review. We totally agree with moving T-cell lymphoma and mantle cell lymphoma into the aggressive lymphoma section (2.1.2. Mantle cell lymphoma (MCL), lines 254-279; 2.1.3. T-cell lymphoma, lines 281-298). We have followed the suggestion of Reviewer 2 and included a paragraph on prognostic factors/biomarkers for patients with DLBCL/HGL and added the following: “A range of factors may affect CAR-T cell therapy efficacy, including patient and disease characteristics, CAR-T cell manufacturing and the type and depth of lymphodepletion. Attempts to identify molecular biomarker of response to CAR T cell therapy (e.g. tumour expression of CD19, CD3, PD-1, PD-L1, CD3, TIM3 and LAG3) have so far been disappointing[6]; however, in the era of precision medicine, identifying patients more likely to respond to adoptive T-cell therapy and improving prognostic predictions is of paramount importance and should be prioritized for future trials. ”